# Development of Fermented Cricket Paste and Its Characteristic Comparison with Traditional Fermented Shrimp Paste (*terasi*)

**DOI:** 10.3390/foods14203562

**Published:** 2025-10-19

**Authors:** Reggie Surya, Felicia Tedjakusuma, Kantiya Petsong, Aphinya Thinthasit, David Nugroho

**Affiliations:** 1Food Technology Department, Faculty of Engineering, Bina Nusantara University, Jakarta 11480, Indonesia; 2Department of Food Science and Technology, Faculty of Agro-Industry, Kasetsart University, Bangkok 10900, Thailand; 3Department of Food Technology, Faculty of Technology, Khon Kaen University, Khon Kaen 40002, Thailand; 4Department of Integrated Science, Faculty of Science, Khon Kaen University, Khon Kaen 40002, Thailand

**Keywords:** *Acheta domesticus*, edible insect, fermentation, Indonesia, shrimp paste

## Abstract

The demand for sustainable protein has increased interest in edible insects, and fermentation can improve the sensory and nutritional profiles of novel foods. This study aimed to develop a fermented cricket paste using a method analogous to traditional shrimp paste and evaluate the physiochemical, nutritional, microbiological, and sensory properties. Both pastes were produced via a biphasic fermentation protocol and subsequently analyzed for their physicochemical, nutritional, microbiological, and sensory properties including consumer acceptance testing in a chili paste. The cricket paste showed a comparable protein content, but higher in carbohydrates and lower in fat. It also showed enhanced stability and safety, with a significantly lower level of histamine (2.37 ppm) compared with shrimp paste (50.51 ppm). While the microbial profiles were broadly similar, the cricket paste had a lower lactic acid bacteria community. Sensory analysis revealed distinct profiles, with the cricket paste characterized by a dark color, coarse texture, nutty/earthy aroma, bitter, and less umami. Despite these differences, consumer liking for chili pastes made with either product was statistically identical. Cricket paste is therefore a safe, stable, and nutritionally advantageous alternative that is highly acceptable to consumers when used as a culinary ingredient.

## 1. Introduction

The global food system is under intense strain as the demand for animal-derived protein is projected to continue to increase [1]. Conventional livestock farming, in particular, has been identified as a major contributor to environmental degradation. Furthermore, the production of conventional protein crops for animal feed, most notably soy, is directly linked to widespread deforestation, land degradation, and a loss of biodiversity [2]. This creates an urgent need for sustainable and alternative sources of protein that are nutritious and sustainable.

Among the viable alternatives, the human consumption of insects (entomophagy) has emerged as a highly promising solution [3]. Indeed, edible insects have long been discussed as part of the human diet [4]. In 1975, Meyer-Rochow proposed insects as one avenue to ease food shortages of that era [4]. Today, however, the global nutrition challenge has shifted: while pockets of food insecurity persist, the dominant burden increasingly stems from the overconsumption of energy-dense, nutrient-poor foods, rising obesity, and food overproduction and waste [5,6]. The priority is therefore related to broad access to nutritious, safe, and culturally acceptable foods with a modest environmental footprint. Thus, insect protein could align with this priority when it delivers high-quality protein as well as favorable safety, stability, and sustainability profiles.

Among the vast number of edible insects, house crickets (*Acheta domesticus*) have been identified as a particularly viable protein source due to their high nutritional profile and exceptional production efficiency [7]. Cricket contains a high protein content, ranging from 55% to 73% of their dry matter, which is comparable or superior to conventional meats like beef and chicken [8]. Crickets demonstrate a remarkably high feed conversion efficiency, requiring significantly less feed to produce 1 kg of meat compared with chickens, pigs, and especially cattle [3]. Crickets also require substantially less water and land than cattle. In practice, production is typically ground-level, carried out through converting disused houses or other buildings into rearing facilities with modest investments in temperature/humidity control, shelving, and biosecurity [9,10]. As non-ruminants, crickets do not generate enteric methane, and their overall greenhouse gas emissions per kilogram of protein are markedly lower than those of conventional livestock [11,12].

Despite these benefits, the adoption of entomophagy, especially in Western cultures, is hindered by psychological barriers like food neophobia [13]. A strategy to overcome this is processing insects into non-recognizable forms, such as pastes or powders, and incorporating them as ingredients in familiar composite foods (e.g., sauces, spreads, or seasonings). Fermentation has been practiced for a thousand years in Southeast Asia [14]. Fermentation could not only enhance shelf-life and nutrient bioavailability, but also enrich sensory characteristics by developing complex and appealing flavor profiles, particularly the savory umami taste that arises from the microbial breakdown of proteins into free amino acids such as glutamate [15]. This sensory transformation can make novel insect-based ingredients more palatable and acceptable to consumers unfamiliar with them.

*Terasi* is a traditional fermented shrimp paste from Indonesia [16]. It is traditionally made by combining marine planktonic shrimp (*Acetes indicus*), locally known as *udang rebon*, with salt. *Terasi* is widely utilized in Indonesian cooking, either as a seasoning to enrich flavor or as a key ingredient in *sambal terasi*, the traditional chili-based condiment [17]. Throughout Asian countries, similar products to *terasi* exist under different local names such as *belacan* in Malaysia, *kapi* in Thailand, *mam tom* in Vietnam, and *bagoong alamang* in the Philippines [18].

Due to its profound cultural significance and well-defined sensory profile, *terasi* serves as an ideal traditional benchmark for the development and characterization of a new fermented protein paste. The underlying principle of using salt and microbial activity to develop shrimp paste can be directly applied to crickets. The utilization of protein-rich crickets as a raw material for a fermented paste could serve as a sustainable and high-protein alternative to traditional shrimp pastes. However, the feasibility of producing a cricket-based ferment that mimics the complex physicochemical and sensory properties of traditional shrimp paste remains largely unexplored.

Therefore, this study aimed to develop a novel fermented cricket paste using a method analogous to that of traditional *terasi* production. The primary objective was to comprehensively evaluate and compare the key characteristics of this new product with those of conventional fermented shrimp paste. The comparison encompassed the physicochemical properties, nutritional composition, microbiology profile, and a sensory profile to determine its potential as a viable and sustainable substitute for traditional *terasi*.

## 2. Materials and Methods

### 2.1. Preparation of Cricket Paste

The preparation of cricket was performed following the method previously described by Tedjakusuma et al. [19]. Briefly, wingless live house crickets (*Acheta domesticus*) in the juvenile stage (~3 week old post-hatch, sex-mixed), clinically healthy, and free of visible defects, were obtained from a commercial farm. Crickets were held off-feed for 48 h with access to moisture only prior to processing to purge the gut contents. Live crickets were humanely euthanized by rapid cryo-immersion in liquid nitrogen (−196 °C; 60–120 s) using a perforated stainless cup. Specimens were then thawed on ice and immediately rinsed and blanched for 5 min. Then, crickets were prepared for paste fermentation according to the method of shrimp paste fermentation described by Surya et al. [18]. Drained crickets were then mixed with salt to a final concentration of 15% (*w*/*w*). A biphasic fermentation protocol was used. The primary fermentation stage involved incubating the salted crickets in a hermetically sealed container for 48 h at 25 °C. Following this, the mixture was homogenized into a paste, formed into discs approximately 8–10 cm in diameter, and subjected to convective drying in an oven at 50 °C for 4 h to reduce the water activity. For the secondary fermentation or maturation phase, these paste discs were aged under ambient conditions for 90 days. This maturation period was selected to reflect traditional production and consumption timelines. Shrimp paste was prepared from planktonic shrimp (*Acetes indicus*) bought from a local market, using the same method and conditions as described by Surya et al. [18].

### 2.2. Microbial Load Analysis

The total number of viable bacteria was assessed at the end of secondary fermentation (maturation) for 90 days following the protocol described by Pongsetkul et al. [20], using plate count agar supplemented with 10% NaCl (pH 7.5). Tenfold dilutions were spread onto agar plates and incubated at 35 °C for 5 days. Lactic acid bacteria were quantified using de Man, Rogosa, and Sharpe (MRS) agar supplemented, followed by incubation at 35 °C for 5 days [21]. Halophilic bacteria were measured using JCM medium followed by incubation at 35 °C for 5 days [22].

### 2.3. Proximate Analysis

Proximate composition was determined following the standard procedures of the Association of Official Analytical Chemists (AOAC) and the American Oil Chemists’ Society (AOCS), with slight modifications [23]. Moisture content was quantified gravimetrically after oven-drying the samples at 135 °C for 2 h, while the ash content was measured by gravimetric analysis after calcination at 550 °C. Lipid content was determined using Soxhlet extraction with petroleum ether (40–60 °C), and the protein content was assessed via the Kjeldahl method. Carbohydrate levels were calculated by difference.

### 2.4. Physiochemical Analysis

#### 2.4.1. Physiochemical Properties

Color was measured using a colorimeter (NH310 3nh, China) based on the CIE Lab* system. Measurements were taken at three different points on each sample, and the mean values of L* (lightness), a* (greenness/redness), and b* (yellowness/blueness) were recorded [24]. Water activity was measured with a water activity analyzer (AwTherm, Rotronic, Bassersdorf, Germany). The pH was determined using a pH meter (Type 766, Knick International, Berlin, Germany) as described by Nirmal and Benjakul [25].

#### 2.4.2. In Vitro Digestibility

In vitro protein digestibility was determined following the method of Surya et al. [26]. Briefly, 5 mL of sample was hydrolyzed with a protease mixture of trypsin, chymotrypsin, and pancreatin at 37 °C for 30 min. A blank (without enzymes) and casein control were included. The reaction was terminated, and undigested proteins were precipitated with 2 mL of 0.5 M trichloroacetic acid before being separated by centrifugation. The concentration of soluble peptides in the resulting supernatant was then determined spectrophotometrically at 750 nm using the Folin–Ciocalteu reagent. A blank (without enzymes) and a casein control were analyzed in parallel.

In vitro lipid digestibility was analyzed according to the modified titration method of Lie and Surya [27]. Briefly, 25 mL samples were hydrolyzed with lipase (10 mg), pancreatin (10 mg), and bile salt (20 mg) at 37 °C for 15 min. Subsequently, free fatty acid (FFA) of the digestate was measured via Formol titration. The sample was prepared with 1.35 mL of 37.5% formaldehyde and phenolphthalein, then titrated against 0.1 N sodium hydroxide (NaOH). Lipid digestibility was calculated from the FFA content and reported as a percentage of oleic acid equivalents.

#### 2.4.3. Trichloroacetic Acid (TCA)-Soluble Peptide, Total Volatile Base Nitrogen (TVB-N), Thiobarbituric Acid Reactive Substances (TBARS), Reducing Sugar, Histamine, and Acrylamide Assays

The degree of hydrolysis (DH) of proteins, expressed as the proportion of peptide bonds cleaved in a hydrolysate, was assessed through the quantification of TCA-soluble peptides. The analysis followed the method of Pongsetkul et al. [23], where 3 g of ground sample was homogenized with 27 mL of cold 5% TCA (11,000 rpm, 1 min), kept on ice for 30 min, and centrifuged at 5000× *g* for 20 min at 4 °C. Soluble peptides in the supernatant were then measured using the Lowry method by UV–Vis spectrophotometry (Spectroquant Prove 100, EMD Millipore, MA, USA) at 650 nm. Results were expressed as mmol tyrosine equivalent/g dry sample.

TVB-N contents were determined using Conway’s micro-diffusion method as described by Rujirapong et al. [28]. Briefly, 8 g of sample was extracted with 2 mL of 4% trichloroacetic acid (TCA) and the mixture was filtered through Whatman No. 41 paper. The filtrate was then diluted with 4% TCA to a final volume of 10 mL. Aliquots of the filtrate were placed in the outer ring of a Conway unit, while boric acid containing an indicator was pipetted into the inner ring. Saturated K_2_CO_3_ was added to the outer ring opposite the sample, and the unit was rotated to mix the reagents. The plate was incubated at room temperature for 3 h, after which the inner ring was titrated with 0.02 N HCl until the original indicator color was restored. A blank was prepared using TCA solution in place of the sample. TVB-N was expressed as mg N/100 g.

Lipid oxidation was evaluated via the TBARS assay according to Nirmal and Benjakul [25]. Briefly, 1 g of sample was mixed with 9 mL of 0.375% TBA solution. The mixture was heated in boiling water for 10 min, cooled, and centrifuged at 4000× *g* for 20 min. The absorbance of the supernatant was determined at 532 nm (Spectroquant Prove 100, EMD Millipore, MA, USA). TBARS values were calculated against a malondialdehyde (MDA) standard curve (0–5 ppm) and expressed as mg MDA/kg dry sample.

The reducing sugar content was quantified using the 3,5-dinitrosalicylic acid (DNSA) assay, following the procedure described by Krivorotova and Sereikaite [29] with minor modifications. To prepare the DNSA reagent, 1 g of DNSA and 30 g of sodium–potassium tartrate were dissolved in 80 mL of 0.5 N NaOH at 45 °C. After complete dissolution, the solution was cooled to room temperature, and the final volume was adjusted to 100 mL with distilled water. For analysis, 2 mL of DNSA reagent was mixed with 1 mL of the sample (1 mg/mL) and heated at 95 °C for 5 min. The mixture was then cooled, diluted with 7 mL of distilled water, and its absorbance measured at 540 nm using a UV–VIS spectrophotometer (Spectroquant Prove 100, EMD Millipore, MA, USA). Reducing sugar was quantified against a calibration curve constructed with D-glucose standards (200–1000 mg/L) and expressed as milligrams of D-glucose equivalent per gram (mg GE/g).

Histamine was analyzed using a commercial histamine assay kit (Megazyme, Dublin, Ireland). Briefly, 2 g of sample was homogenized with 15 mL of 100 mM EDTA buffer (pH 8). The mixture was boiled and centrifuged at 10,000× *g* for 5 min. The supernatant was reacted with histamine dehydrogenase (HDH), which reduced iodonitetrazolium chloride (INT) to a formazan dye. Histamine concentration was then measured spectrophotometrically at 492 nm (Spectroquant Prove 100, EMD Millipore, MA, USA) and expressed as mg histamine/kg dry sample (ppm).

Acrylamide was determined using an Acrylamide-ES ELISA Kit (Life Technologies, Delhi, India) following the manufacturer’s protocol. Briefly, samples were incubated with acrylamide-linked antibody solution at 4 °C for 60 min, washed, and subsequently incubated with substrate solution for 30 min at room temperature. Absorbance was recorded at 450 nm with a microplate ELISA reader (Infinite 200 PRO, Tecan, Männedorf, Switzerland). Results were expressed as μg acrylamide/kg dry sample (ppb).

### 2.5. Preparation of Cricket Chili Paste

Cricket chili paste was prepared according to Surya et al. [18] with slight modifications. Briefly, cayenne peppers, garlic, and shallots were stir-fried in palm oil for 5 min. Separately, cricket paste was toasted in a heated pan for 2 min to enhance its aroma. The toasted cricket paste was then mixed with the stir-fried ingredients and granulated sugar until a smooth paste was obtained. The mixture was boiled, cooled, and stored at 4 °C before analysis.

### 2.6. Sensory Evaluation

#### 2.6.1. Discriminative Test (Triangle Test)

A triangle test was performed to determine whether a perceivable sensory difference existed between the cricket and shrimp products. A panel of 84 participants was presented with three samples (two identical, one different) and asked to identify the “odd” sample for both the plain pastes and the chili pastes.

#### 2.6.2. Descriptive Intensity Rating

A total of 172 panelists were asked to rate the intensity of the cricket paste and cricket chili paste. Various attributes, such as color, texture, and aroma/flavor notes, were evaluated on a 9-point scale, where 1 was “much weaker” and 9 was “much stronger” relative to the shrimp paste, which was used as a control reference normalized to a score of 5.

#### 2.6.3. Hedonic Rating

A total of 172 untrained panelists were asked to evaluate five attributes of cricket and shrimp chili paste including color, texture, aroma, taste, and overall acceptability. The consumer acceptance of the chili pastes was measured using a 9-point hedonic scale (1 = dislike extremely, 9 = like extremely).

### 2.7. Statistical Analysis

All analytical experiments were performed in triplicate (*n* = 3), and the results were reported as the mean ± standard deviation. Sensory data were collected from 84 or 172 panelists as specified. Statistical significance between the samples for each parameter was determined using the *t*-test, with differences considered significant at a *p*-value of less than 0.05 (*p* < 0.05).

## 3. Results

### 3.1. Physiochemical Characteristics of Cricket Paste

Table 1 shows the physicochemical analysis between the cricket and shrimp pastes that influence their quality, stability, and safety. Color analysis showed that cricket paste was found to be significantly darker when compared with the shrimp paste. This is likely attributable to a more extensive Maillard reaction occurring within the cricket paste matrix. The Maillard reaction is a chemical reaction between amino acid residues and reducing sugars that leads to the formation of browning that contributes to the flavor and color of foods [30]. This hypothesis is strongly supported by the chemical analysis, which showed that the cricket paste contained a significantly higher concentration of reducing sugars (16.69 mg/g) than the shrimp paste (9.42 mg/g). Further evidence for a more pronounced Maillard reaction in the cricket paste is its higher level of acrylamide (52.76 ppb) compared with the shrimp paste (41.19 ppb), as acrylamide is a known by-product of this specific chemical pathway. Therefore, the higher availability of reducing sugars in cricket paste likely accelerated the Maillard reaction, leading to its dark color and potentially influencing its aroma and flavor.

Cricket paste was significantly lower in pH (6.3) and water activity (a_w_ = 0.74) than shrimp paste (pH 6.5, a_w_ = 0.78), therefore creating a less favorable environment for the growth of spoilage microorganisms and pathogens. Water activity is an important parameter for food preservation, as it measures the amount of free water available for microbial metabolism [31]. Most spoilage bacteria require an a_w_ value of 0.90 or higher to grow [32]. The a_w_ values indicate that both pastes are categorized as intermediate moisture foods (IMF). This enhanced stability is further evidenced by chemical markers of degradation. Cricket paste showed significantly lower levels of total volatile base nitrogen (TVB-N) and thiobarbiturate acid reactive substances (TBARS) compared with shrimp paste. TVB-N is an indicator of protein degradation by microbial enzymes [33], while TBARS measures the extent of lipid oxidation, which leads to rancidity and off-flavors [34]. Lower values of TVB-N and TBARS in cricket paste indicate a more stable product and are more resistant to proteolysis and lipid oxidation over time.

In addition, cricket paste also showed a much lower concentration of TCA-soluble peptides, which indicated a lower rate of protein hydrolysis. From a food safety perspective, the cricket paste contained much lower concentrations of histamine (2.37 ppm) compared with shrimp paste (50.51 ppm), which highlights another advantage of insect-based products. Histamine is a biogenic amine formed from the degradation of histidine by microbes. Histamine is a significant safety concern in fermented seafood products and can cause scombroid poisoning [35,36]. According to the U.S. Food and Drug Administration (FDA), fish and fishery products containing histamine levels of 35 ppm or higher should be adulterated, while those containing 200 ppm or higher are considered harmful to human health [37]. Therefore, the histamine levels in cricket paste are not only within these safe limits, but also indicate the potential for a safer alternative to traditional fishery products that are susceptible to histamine formation.

### 3.2. Nutritional Characteristics of Cricket Paste

The proximate composition of the cricket and shrimp pastes are shown in Table 2. Cricket paste showed a significantly higher crude protein content (77.58%) than shrimp paste (72.46%), however, the chitin-corrected protein content appeared to be statistically indifferent between the cricket paste and shrimp paste. Cricket paste also had a lower lipid content of 1.39% (*w*/*w*), whereas the shrimp paste contained 8.36% (*w*/*w*), resulting in a significantly higher protein-to-lipid ratio for the cricket paste. Furthermore, the carbohydrate fraction, obtained by difference, was significantly higher in the cricket paste (18.47%) relative to the shrimp paste (11.13%), a finding likely attributable to structural polysaccharides such as chitin from the insect exoskeleton [38]. The ash/mineral content, however, did not differ significantly between the samples. The in vitro protein digestibility of cricket paste was significantly lower at 69.67% compared with 80.41% for the shrimp paste. A higher digestibility indicates greater susceptibility to proteolytic enzymes, which translates to a more efficient bioavailability of essential amino acids for the body.

### 3.3. Microbiological Profile of Cricket Paste

The microbial counts and community profiles of cricket paste and shrimp paste are presented in Figure 1 and Figure 2. Both products exhibited high total viable counts and comparable populations of halophilic bacteria, consistent with the characteristics of high-salt fermented foods. In contrast, the mold and yeast populations were low and did not differ significantly between samples, suggesting that the fermentation conditions effectively suppressed fungal growth. The most notable difference was observed in the lactic acid bacteria (LAB) population, which was significantly lower in cricket paste than in shrimp paste. This quantitative variation was supported by qualitative community differences. At the phylum level, both pastes were dominated by Firmicutes; however, shrimp paste also contained a higher relative abundance of Tenericutes. At the genus level, halophilic fermenters such as *Halanaerobium* and *Tetragenococcus* were higher in shrimp paste than cricket paste. These microbial distinctions were associated with reduced proteolysis and deamination in cricket paste, as indicated by its lower levels of TCA-soluble peptides and TVB-N. Such differences can be attributed to multiple factors that influence LAB populations in fermented foods, particularly the fermentation conditions, native microbiota of raw materials, and the chemical composition of the food matrix [39,40].

### 3.4. Sensory Characteristics of Cricket Paste

The triangle test confirmed that panelists could reliably distinguish between the cricket and shrimp pastes, both in their base form and when made into a chili paste (Table 3). This indicates a clear and noticeable difference in their sensory profiles. The descriptive analysis in Figure 3 further detailed these distinctions, showing that the cricket paste was perceived as significantly darker and possessed a coarser, grittier texture while being less glossy than the shrimp paste baseline. In terms of aroma, cricket paste had a significantly weaker fishy aroma and was instead defined by a dominant and intense nutty and earthy aroma. It was also rated as having a stronger acidic aroma, with a fermented aroma intensity comparable to that of shrimp paste. These differences were also found in the chili paste form. The resulting cricket chili paste was rated as significantly less salty and possessed a weaker umami intensity than shrimp chili paste alongside a slightly more pronounced bitterness and a comparable level of sweetness. These sensory perceptions aligned well with the physicochemical data, where the darker color corresponded to the lower L* value, and the weaker umami intensity was consistent with the lower concentration of flavor-active TCA-soluble peptides.

Crucially, despite these distinct profiles, there was no significant difference in consumer liking between the cricket chili paste and shrimp chili paste across all attributes (Table 4). This finding is important, as it demonstrates that cricket paste is a highly acceptable substitute when used as an ingredient in a final product. This aligns with the common use of shrimp paste (*terasi*), which is rarely consumed alone but is instead a key flavoring ingredient in dishes like sambal. These sensory results indicate a strong potential for the market acceptance of cricket paste as a novel and sustainable food ingredient.

## 4. Discussion

Our findings showed clear, multidimensional differences between the fermented shrimp paste and the newly developed fermented cricket paste. Relative to the shrimp paste, the cricket paste exhibited a leaner proximate profile and more favorable indicators of freshness and oxidative stability, alongside markedly low histamine. The physicochemical parameters showed that cricket paste was darker and slightly more acidic, with a lower water activity compared with shrimp paste. Microbiological profiles also diverged, suggesting distinct proteolytic trajectories and flavor-development pathways during ripening.

Physicochemically, the fermented cricket paste and shrimp paste diverged across the color, composition, and stability markers. The cricket paste was markedly darker (lower L* and higher browning index), consistent with stronger Maillard progression driven by its higher reducing-sugar activity; this aligns with the elevated acrylamide signal observed in the insect matrix. It also showed a slightly lower pH and water activity, conditions that favor shelf stability in high-salt ferments. Volatile-base nitrogen (TVB-N) and histamine were also lower in the cricket paste (≈2–3 mg/kg) compared with the shrimp paste (≈50 mg/kg), providing a wide safety margin under the same processing conditions. The markedly lower accumulation likely reflects the insect matrix’s lower free-histidine availability, faster acidification, and reduced water activity as well as a salt-tolerant microbiota that is not dominated by histamine-forming bacteria. Together, these features suggest that, relative to shrimp paste, the cricket paste develops a darker, more acidified, and drier matrix with improved oxidative and chemical stability while exhibiting a distinct pattern of proteolysis and Maillard-derived products.

Proximally, the cricket paste presented a higher crude protein and substantially lower fat than shrimp paste, a leaner profile that corresponds to its significantly lower lipid oxidation (TBARS). A methodological consideration is the likely overestimation of protein in the cricket paste when using Kjeldahl. Because chitin and related non-protein nitrogen from the cuticle can contribute substantially to the total nitrogen, a meaningful share of Kjeldahl-detected N may not originate from true protein [41]. To improve the accuracy, we reported both the crude protein and chitin-corrected protein calculated in the samples following removal of the cricket and shrimp exoskeleton (Table 2), thus suggesting insect-appropriate nitrogen-to-protein conversion factors lower than 6.25 (5.43 and 5.63 for cricket paste and shrimp paste, respectively). These conversion factors were close to the one previously reported to correct the chitin contribution in the total N in yellow mealworm, which was 5.66 [42]. Such a correction would yield a more rigorous nutritional interpretation without diminishing the observed functional digestibility. Consistent with this, the carbohydrate (including chitin) fraction was greater in the cricket paste than in the shrimp paste.

Both fermented cricket paste and shrimp paste are significant protein sources. Their protein concentrations were comparable after adjusting for chitin, while cricket paste showed a higher in vitro protein digestibility, indicating stronger functional protein availability. The protein digestibility of cricket paste could be further improved by applying lactic acid fermentation, as reported in previous studies [43].

Economically, crickets can deliver a similar protein payload at often lower per-kg cost than planktonic shrimp (*udang rebon*) in Indonesian markets. Based on our observation in local settings, the producer-level cricket price was around USD 3–6 per kg versus the planktonic shrimp price commonly set at USD 5–9 per kg. However, since these pastes are used primarily as seasonings (mg-scale per serving), the difference in raw-material price and even total protein per kilogram would have a limited impact on the daily protein intake at the consumer level. The economic benefit is more meaningful for producers and in large-batch food service than for individual nutrition.

Microbiologically, both pastes exhibited similar high-salt ferment communities, with the only significant compositional difference being lactic acid bacteria (LAB), which were relatively less abundant in the cricket paste in our data. In salted seafood ferments, halotolerant LAB, especially *Tetragenococcus*, often emerge under high NaCl and can shape acidification and amine control, although their levels depend on fermentable-carbohydrate availability (reported to be limited in *terasi*) and process conditions [44,45]. The LAB detected in our pastes plausibly derive from two routes: (i) carryover from the raw substrates’ gut microbiota, as LAB have been isolated from penaeid shrimp intestines and are common in insect guts (e.g., *Pediococcus*/*Lactobacillus*/*Lactiplantibacillus* in house crickets), and (ii) environmental sources selected during curing and maturation [46,47]. The relatively lower LAB in the cricket paste may therefore reflect substrate differences (less readily fermentable carbohydrates), salt/pH regimes that favor other halophiles over insect-derived LAB, or the absence of strongly halotolerant LAB strains. Such patterns are consistent with prior observations across shrimp and fish-sauce fermentations [48].

Regarding digestibility, matrix effects are considered as an essential matter. While our data indicated higher in-vitro protein digestibility for the cricket paste than for shrimp paste, further improvements are plausible. Prior work with cricket flour shows that lactic acid fermentation can enrich free amino acids and beneficially modify the composition; *Lactiplantibacillus*/*Lactobacillus* fermentations in insect flours and sourdough systems are known to increase proteolysis, which is often associated with better in vitro digestibility [49]. Translating this to a paste system suggests employing salt-tolerant LAB starters (e.g., *Tetragenococcus* spp.) or a staged process (primary high-salt ripening followed by a controlled finishing step), approaches that have improved amino acid profiles and even reduced biogenic amines in salt-fermented fish sauces [50,51].

Finally, sensory results help contextualize the contrasts between cricket paste and shrimp paste. Although panelists were readily able to discriminate the two products, hedonic ratings were comparable when both were used as a seasoning in a familiar application (chili paste). This indicates that the application context can mitigate unfamiliar flavor notes and support the culinary viability of insect-based ferments. Process optimization via salt level, temperature–time profiles, and starter choice may further modulate umami, suppress bitterness, and align flavor with consumer expectations.

Overall, fermented cricket paste emerged as a credible, application-ready alternative to shrimp paste, with a pathway to strengthen its nutritional claims (true-protein reporting) and further elevate digestibility and flavor through targeted LAB-assisted process design.

## 5. Conclusions

This study demonstrated that cricket paste is a viable and advantageous alternative to traditional shrimp paste in terms of nutrition, safety, and self-stability while maintaining high consumer acceptance in a final product application. The cricket paste showed a comparable protein content and a significantly lower fat content compared with shrimp paste. Furthermore, it exhibited enhanced physicochemical stability and a significant food safety advantage. While microbiological analyses revealed compositionally similar ecosystems, the cricket paste was shown to have a smaller but more metabolically potent community of lactic acid bacteria, which resulted in its unique sensory and chemical characteristics. Despite these discernible differences, the hedonic sensory analysis showed no significant difference in consumer liking between chili pastes made with either cricket or shrimp paste. This equivalency in consumer acceptance is the most critical finding, indicating that cricket paste can successfully substitute shrimp paste in culinary applications without compromising consumer preference. Therefore, cricket paste represents a promising, safe, and nutritious ingredient with high potential for market adoption as a sustainable alternative in the food industry.

## Figures and Tables

**Figure 1 foods-14-03562-f001:**
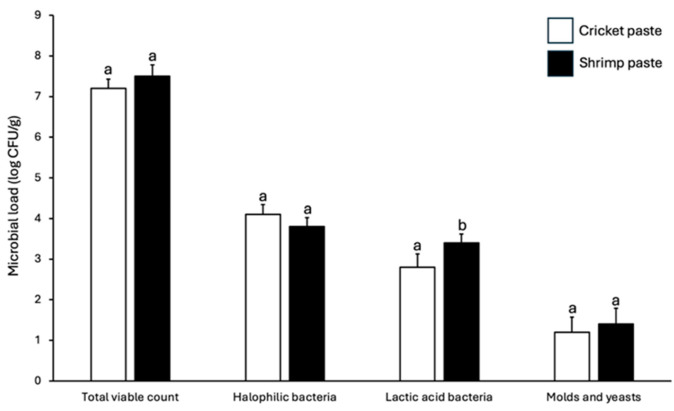
Microbial load in cricket paste and shrimp paste. Data (*n* = 3) were expressed as the mean ± SD. Different letters in a group indicate a significant difference (*p* < 0.05).

**Figure 2 foods-14-03562-f002:**
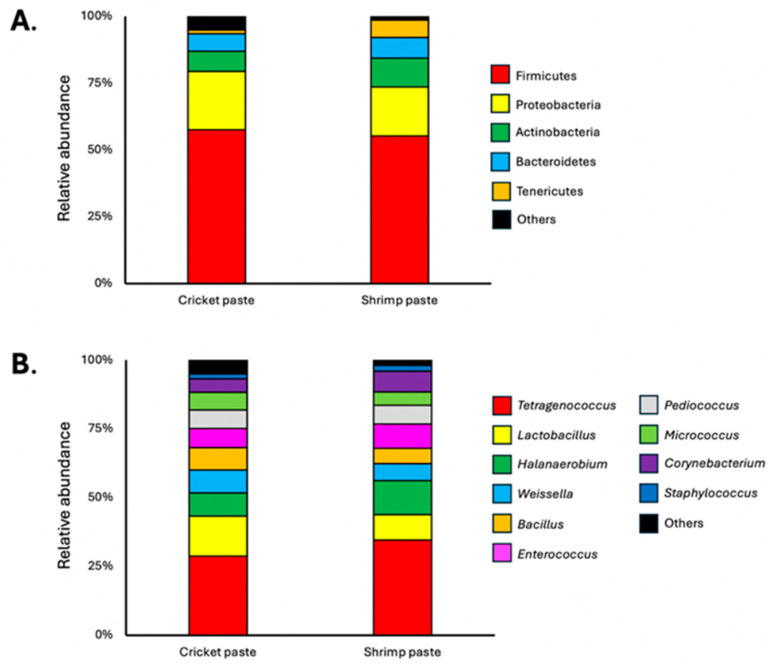
Relative microbiological abundance of bacteria at the (**A**) phylum level and (**B**) genus level in cricket paste and shrimp paste (*n* = 3).

**Figure 3 foods-14-03562-f003:**
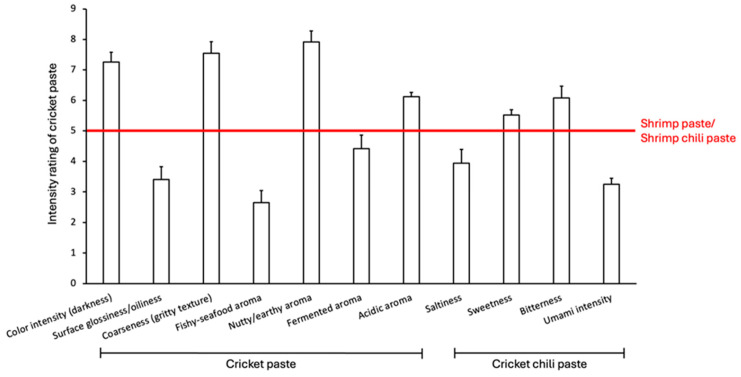
Intensity of various sensory parameters of cricket paste and cricket chili paste assessed by the panelists (*n* = 172) relative to the shrimp paste/shrimp chili paste, whose intensity of all parameters was normalized to 5, as indicated by the red line. The intensity scale ranged from 1 (much weaker than shrimp paste) to 9 (much stronger than shrimp paste).

**Table 1 foods-14-03562-t001:** Physicochemical analysis of cricket paste and shrimp paste.

	Unit	Cricket Paste	Shrimp Paste
Color analysis			
L* (lightness)	-	32.5 ± 2.1 ^b^	42.7 ± 1.8 ^a^
a* (red-green)	-	7.3 ± 0.29 ^b^	12.3 ± 0.6 ^a^
b* (yellow-blue)	-	14.8 ± 1.2 ^b^	9.6 ± 0.7 ^a^
ΔE	-	9.9 ± 1.5	0
Chemical analysis			
pH	-	6.3 ± 0.1 ^b^	6.5 ± 0.1 ^a^
Water activity (a_w_)	-	0.74 ± 0.01 ^b^	0.78 ± 0.01 ^a^
Moisture content	%*w*/*w*	42.2 ± 1.5 ^a^	48.4 ± 1.7 ^b^
TCA-soluble peptides	mg tyrosine equivalent/g	8.62 ± 0.49 ^b^	12.15 ± 0.58 ^a^
Total volatile base nitrogen	mg N/100 g	80.61 ± 2.13 ^b^	92.74 ± 4.23 ^a^
TBARS	ppm MDA	1.94 ± 0.13 ^b^	3.27 ± 0.15 ^a^
Reducing sugars	mg GE/g	16.69 ± 1.09 ^a^	9.42 ± 0.77 ^b^
Histamine	ppm	2.37 ± 0.09 ^b^	50.51 ± 1.65 ^a^
Acrylamide	ppb	52.76 ± 5.64 ^b^	41.19 ± 3.83 ^a^

Data (*n* = 3) are reported as the mean ± SD. Different letters in a line indicate a significant difference among samples (*p* < 0.05).

**Table 2 foods-14-03562-t002:** Nutritional proximate composition and in vitro nutrient digestibility of cricket paste and shrimp paste.

	Unit	Cricket Paste	Shrimp Paste
Ash/minerals	%*w*/*w*	12.76 ± 3.85 ^a^	15.22 ± 4.41 ^a^
Protein (crude)	%*w*/*w*	77.58 ± 3.17 ^b^	72.46 ± 2.82 ^a^
Protein (chitin-corrected)	%*w*/*w*	67.38 ± 2.58 ^a^	65.29 ± 2.19 ^a^
Fat	%*w*/*w*	1.39 ± 0.22 ^a^	8.36 ± 0.75 ^b^
Carbohydrate (including chitin)	%*w*/*w*	18.47 ± 1.27 ^b^	11.13 ± 0.89 ^a^
Protein digestibility	%	80.41 ± 6.13 ^b^	69.67 ± 4.19 ^a^
Lipid digestibility	%	71.15 ± 5.33 ^a^	74.26 ± 4.65 ^a^

Data (*n* = 3) are reported as the mean ± SD. Different letters in a line indicate a significant difference among samples (*p* < 0.05).

**Table 3 foods-14-03562-t003:** Results of the discriminative sensory test (triangle test).

Sample	No. of Panelists	No. of Correct Judgments to Achieve (α = 0.05, *p* = 1/3)	No. of Correct Judgements
Cricket paste vs. shrimp paste	84	36	62
Cricket chili paste vs. shrimp chili paste	84	36	49

**Table 4 foods-14-03562-t004:** Sensory acceptance of cricket chili paste and shrimp chili paste measuring the panelists’ liking expressed in hedonic rate (scale 1–9) and rank.

	Cricket Chili Paste	Shrimp Chili Paste
Appearance	7.16 ± 0.95 ^a^	7.22 ± 0.83 ^a^
Aroma	7.28 ± 1.27 ^a^	7.49 ± 1.17 ^a^
Taste	6.95 ± 0.92 ^a^	7.26 ± 0.85 ^a^
Overall rate	7.19 ± 1.39 ^a^	7.36 ± 1.08 ^a^
Overall rank	1.58 ± 0.13 ^a^	1.42 ± 0.17 ^a^

Data (*n* = 172) are reported as the mean ± SD. Different letters in a line indicate a significant difference among samples (*p* < 0.05).

## Data Availability

The original contributions presented in this study are included in the article. Further inquiries can be directed to the corresponding authors.

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
