# Peer review of "Development of Fermented Cricket Paste and Its Characteristic Comparison with Traditional Fermented Shrimp Paste (terasi)"

_foods, 2025, doi:10.3390/foods14203562_

Round 1

Reviewer 1 Report

Comments and Suggestions for Authors

Title:  “Development of fermented cricket paste….

Authors: Reggie Surya et al.

This is an excellent, very professionally conducted and beautifully presented study and I have few comments. The first short paragraph of the Introduction, however, does require some clarification. When it was first suggested by V.B. Meyer-Rochow in 1975  (a paper that should have been cited: 1975 “Can insects help to ease the problem of world food shortage?” Search 6(7), 261-263) pointing out  that edible insects could help reducing food shortages, it was a different era. Today (despite a few regions with poor food security, such as Tigray in Northeastern Ethiopia) there is nowhere in the world blatant starvation. But there is, however, a serious problem of obesity, overeating and food overproduction , see  https://www.bbc.com/news/health-68436642 .One billion people are obese globally: https://data.worldobesity.org/rankings/ , with significantly more people dying from the consequences of too much rather than too little food!  See also the Editorial in  “Frontiers in Physiology” 2024, “Novel strategies targeting obesity and metabolic diseases” by Xinran Ma et al. Doi: 10.3389/fphys.2024.1342943.

The problem the world faces today is therefore a different one from 1975 as not shortage of food per se, but consumption of too much unhealthy food stuff needs to be tackled.  The emphasis has to be on nutritious food stuffs and that fermented crickets and shrimps ARE nutritious, is something  you need to show or stress in your paper. As to the global population: In all European countries and many Asian countries (China recently announced a fall in population and India is expected to reach zero population growth, too)  populations are decreasing: Population projected to decline in two-thirds of EU-region 2021, available online: https://e.europa.eu/eurostat/web/products-eurostat-news/-/ddn-20210430-2

Nuttall, C.N. 2022. Population decline to the emerging Europe back to the early 20th century. (Available online): 

Tsuchiya, H. 2022. East Asia’s looming demographic crisis. https://www.nippon.com/en/in-depth/d00639).

L53: actually there is also  ‘Neophilia’ (the joy of trying something new) and the fact that Europe, too, had a long entomophagous history with Greek and Romans of the Antiquity feasting on various beetle grubs as well as cicadas and recipes to prepare a tasty soup of cockchafers (May or Junebugs) were still available at the end of the 19th century in Germany. However, this is just for information and need not be mentioned in your paper!

What IS important, is to mention the species of cricket that you used in your investigation. There are many species of crickets and several are being reared under controlled conditions. For Acetes indicus write “…the marine planktonic shrimp Acetes indicus…)

Under the heading 2.2:  when was the microbial load assessed?

Under 2.4.1 Physicochemical properties: why was the viscosity of the pastes not determined? Was it not considered to be important (I think it is) or was it technically not possible or difficult?

On Lines 218 and 270 you use ‘compared to’ when you had better use “compared with”!  Explanation: compared to points to similarities (“The climate of Thailand can be compared to that of Malaysia”), but compared with makes it clear that a ‘true comparison’ has occurred.

Finally, was the paper by Elena Bartkiene et al 2023 in the MDPI journal FERMENTATION 9/3, 153, doi: 10.3390/Fermentation9020153 of no use or importance, or why was it not considered to be discussed? There is also a substantial section on fermented food stuffs in doi: 10.3390/foods14193371, which could possibly be of interest.

In summary an excellent paper!

Author Response

We would like to sincerely thank the Editor and Reviewers for their thoughtful and constructive feedback. We have revised the manuscript accordingly and indicate below, point-by-point, how each comment has been addressed. All changes are incorporated in the revised version.

Reviewer 1

  1. “Clarify the first paragraph of the Introduction; cite Meyer-Rochow (1975) and note that today’s problem is obesity/overconsumption/overproduction.”
    Response: Revised. We now cite Meyer-Rochow’s 1975 proposal, and explicitly frame today’s nutrition challenge as dominated by overconsumption of energy-dense, nutrient-poor foods, rising obesity, and food overproduction/waste, with supporting references (Line 41-50).
  2. “Specify the species of cricket; clarify phrasing for Acetes indicus.”
    Response: Done. We specify house crickets (Acheta domesticus) in Line 51 and marine planktonic shrimp (Acetes indicus) in Line 74.
  3. “When was microbial load assessed?”
    Response: Added. Total viable counts, LAB, and halophiles were assessed at the end of secondary fermentation (maturation) for 90 days, as indicated in Line 116-117.
  4. “Why no viscosity measurement?”
    Response: Even though we called it paste, originally the products (terasi, Indonesian shrimp pastes) are in the form of solidified cubes, for which apparent viscosity is not applicable.
  5. “Use ‘compared with’ rather than ‘compared to’ (lines 218, 270).”
    Response: Corrected throughout.
  6. “Discuss Bartkiene et al. (2023) and related fermented-insect work.”
    Response: Added in the Discussion where we connect LAB-assisted fermentation of cricket flours to potential finishing steps for salted pastes (Line 401-403).

Reviewer 2 Report

Comments and Suggestions for Authors

Edible insects are a hot research topic. This study is interesting. There are still some issues in the manuscript that need to be supplemented or revised.

  1. Secondary headings should be numbered sequentially.
  2. In 2.1 and 2.5 section, if necessary, add a flowchart to help readers better understand.
  3. The high level of protein content in insects for food and feed is overestimated. The protein content assessed via the Kjeldahl methodmust be thoroughly discussed.

See:

https://doi.org/10.1016/j.ijbiomac.2024.132787

https://doi.org/10.1016/j.jfca.2017.06.004

  1. Food safety standards vary across different countries and regions. In addition to histamine, it is preferable to supplement test data for other biogenic amines.
  2. A separate discussion section should be included for more thorough analysis.

Author Response

We sincerely thank the Editor and Reviewers for their thoughtful and constructive feedback. We have revised the manuscript accordingly and indicate below, point-by-point, how each comment has been addressed. All changes are incorporated in the revised version.

Reviewer 2

  1. “Headings should be numbered sequentially.”
    Response: Corrected across Sections 2.1–2.7 and 3.1–3.4.
  2. “Consider a flowchart for 2.1/2.5.”
    Response: We opted not to include a flowchart. The revised text in Methods 2.1 and 2.5 now provides a concise, stepwise description (including pre-treatments, time/temperature, and maturation), which we believe is sufficiently clear without an additional figure.
  3. “Protein may be overestimated by Kjeldahl—discuss and consider corrective approach.”
    Response: Addressed in Results and Discussion. We now report crude protein and chitin-corrected protein in Table 2, as well as added a specific discussion regarding the overestimation (Line 384-398).
  4. “Beyond histamine, consider other biogenic amines.”
    Response: We respectfully maintain that additional biogenic-amine profiling is not required for the scope of this comparative study. In salted fishery/fermented seafood products, histamine is the regulatory sentinel marker. In our samples, both shrimp and cricket histamine were below regulatory concern levels by FDA. To capture the overall extent of protein decomposition, we also quantified total volatile basic nitrogen (TVB-N) using Conway’s micro-diffusion method. Together with lower TBARS and slightly lower pH/aw, these indices indicate limited proteolysis and favorable stability under our high-salt, low-aw maturation regime. Accordingly, we consider histamine monitoring plus TVB-N sufficient for safety/quality assessment in this work
  5. “Add a separate Discussion section.”
    Response: Added as Section 4 (Discussion, Line 363-446).

Reviewer 3 Report

Comments and Suggestions for Authors

Dear authors, you have uploaded the manuscript without the part 4. Discussion. Please improve that at first. Additionally my comments are in the document attached.

Author Response

We sincerely thank the Editor and Reviewers for their thoughtful and constructive feedback. We have revised the manuscript accordingly and indicate below, point-by-point, how each comment has been addressed. All changes are incorporated in the revised version.

Reviewer 3

  1. “A Discussion section is missing.”
    Response: Added as Section 4 (Line 363-446).
  2. “Line 48—crickets are not suited to vertical farming; typical production is at ground level in repurposed buildings.”
    Response: We have revised the Introduction to reflect ground-level production using repurposed/converted buildings with modest environmental controls, shelving, and biosecurity (Line 57-62).
  3. “Line 54—clarify the sentence about overcoming neophobia via processing.”
    Response: Clarified in the introduction section (Line 64-67).
  4. “Lines 55–56—tighten the fermentation/umami sentence.”
    Response: We have revised the sentence (Line 68-71).
  5. “Methods—describe pre-boil handling: starvation and humane euthanasia.”
    Response: Added (Line 96-103).
  6. “Ethics permit for sensory work.”
    Response: Added (Line 469-475).
  7. “Table 2 statistics.”
    Response: We rechecked the analyses and ensured that superscript letters are aligned.
  8. “Microbiological profile—account for effects of pre-processing (starvation). LAB are not common in insect guts; explain observations.”
    Response: We now document the 48 h fasting step in Methodology section, specify that counts were taken after maturation, and explain two plausible LAB sources and the role of salt/pH and fermentable-carbohydrate availability in selecting for halotolerant LAB (Line 412-435).

Round 2

Reviewer 3 Report

Comments and Suggestions for Authors

Dear authors.

I have read your improved version of the manuscript.

The uploaded version is improved (and has a discussion part).

Please read one more time the paper and please correct the latin names of the species / genus (espesially in the discussion part), because  the names should be written in italic.

Other than that minor revision, the manuscript does not require any improvments.
